# A dual-targeted drug inhibits cardiac ryanodine receptor Ca²⁺ leak but activates SERCA2a Ca²⁺ uptake

Jörg W Wegener[1,4] , Gyuzel Y Mitronova[2,4] , Lina ElShareif[1] , Christine Quentin[2] , Vladimir Belov[2], Tatiana Pochechueva[1,4] , Gerd Hasenfuss[1,4] , Lutz Ackermann[3,4], Stephan E Lehnart[1,4]

In the heart, genetic or acquired mishandling of diastolic [Ca²⁺] by ryanodine receptor type 2 (RyR2) overactivity correlates with risks of arrhythmia and sudden cardiac death. Strategies to avoid these risks include decrease of Ca²⁺ release by drugs modulating RyR2 activity or increase in Ca²⁺ uptake by drugs modulating SR Ca²⁺ ATPase (SERCA2a) activity. Here, we combine these strategies by developing experimental compounds that act simultaneously on both processes. Our screening efforts identified the new 1,4-benzothiazepine derivative GM1869 as a promising compound. Consequently, we comparatively studied the effects of the known RyR2 modulators Dantrolene and S36 together with GM1869 on RyR2 and SERCA2a activity in cardiomyocytes from wild type and arrhythmia-susceptible RyR2^R2474S/+ mice by confocal live-cell imaging. All drugs reduced RyR2-mediated Ca²⁺ spark frequency but only GM1869 accelerated SERCA2a-mediated decay of Ca²⁺ transients in murine and human cardiomyocytes. Our data indicate that S36 and GM1869 are more suitable than dantrolene to directly modulate RyR2 activity, especially in RyR2^R2474S/+ mice. Remarkably, GM1869 may represent a new dual-acting lead compound for maintenance of diastolic [Ca²⁺].

## Introduction

During each heartbeat, the intracellular calcium concentration ([Ca²⁺]ᵢ) cycles dynamically between low resting diastolic and high active systolic levels within cardiomyocytes (Bers, 2002; Eisner et al, 2017; Sankaranarayanan et al, 2017): upon electrical excitation, Ca²⁺ influx via voltage-gated Caᵥ1.2 channels activates Ca²⁺-induced Ca²⁺ release via SR ryanodine receptor type 2 (RyR2) Ca²⁺ channels (Fabiato, 1983). This Ca²⁺-induced Ca²⁺ release gives rise to the net systolic [Ca²⁺]ᵢ increase that activates cardiac contraction. For cardiac relaxation, [Ca²⁺]ᵢ is pumped back into the SR Ca²⁺ store by the cardiac SR Ca²⁺-ATPase (SERCA2a) in concert with Ca²⁺ efflux to the extracellular space by the sodium–calcium exchanger (NCX) (McDonough et al, 2002). In healthy conditions, Ca²⁺ influx and efflux and Ca²⁺ release and (re)uptake is strictly balanced to maintain physiological diastolic [Ca²⁺]ᵢ (Walker et al, 2014; Eisner et al, 2020; Hamilton et al, 2021).

In contrast, in various forms of human heart disease, abnormal intracellular Ca²⁺ handling is regarded as one hallmark of disease (Meuse et al, 1992). For example, increased diastolic [Ca²⁺]ᵢ has been observed in muscle strips from failing human hearts (Gwathmey et al, 1988). Subsequently, increased diastolic [Ca²⁺]ᵢ has been confirmed in animal models of heart failure and attributed to diminished SR Ca²⁺ uptake (Hasenfuss et al, 1994; Ribadeau Dumas et al, 1997) and to Ca²⁺ "leak" out of the SR Ca²⁺ store (Louch et al, 2012; Bers, 2014). Increased SR Ca²⁺ leak has been molecularly attributed to RyR2 overactivity mediated by several mechanisms including (i) RyR2 hyper-phosphorylation by PKA because of a chronic hyper-adrenergic state in heart failure (Marx et al, 2000; Potenza et al, 2019), (ii) RyR2 hyper-phosphorylation by Ca²⁺–calmodulin-dependent kinase (CaMKII) involving exchange protein directly activated by cAMP (Epac2) and nitric oxide synthase 1 (Wehrens et al, 2004; Sag et al, 2009; Pereira et al, 2017), and (iii) increased RyR2 oxidation (Andersson et al, 2011; Huang et al, 2021). In addition to these mechanisms of acquired RyR2 channel dys-regulation, missense and truncating mutations in the *RyR2* gene have been linked to increased SR Ca²⁺ leak in transgenic mice carrying human-related RyR2 mutations, for example, R2474S (Lehnart et al, 2008) and P2328S (Lehnart et al, 2004) that mimic the phenotype of catecholaminergic polymorphic ventricular tachycardia (CPVT) (Wleklinski et al, 2020).

Therapeutic strategies for altered intracellular Ca²⁺ handling are mainly focused on modulators of RyR2 activity (Connell et al, 2020; Marks, 2023). Initially, the benzothiazepine derivative JTV519 (K210) was shown to correct SR Ca²⁺ leak in dog samples with heart failure (Yano et al, 2003) and decreased human RyR2 activity in recombinant channels harboring missense mutations (Lehnart et al, 2004). However, JTV-519 also inhibited Ca²⁺-dependent SERCA activity in striated muscle (Darcy et al, 2016). Dantrolene, a

¹Department of Cardiology and Pulmonology, Heart Research Center Göttingen, University Medical Center of Göttingen (UMG), Göttingen, Germany   ²Department of NanoBiophotonics, Max Planck Institute for Multidisciplinary Sciences, Göttingen, Germany   ³Georg-August University of Göttingen, Institute of Organic and Biomolecular Chemistry, Göttingen, Germany   ⁴DZHK (German Centre for Cardiovascular Research), Partner Site Göttingen, Göttingen, Germany

Correspondence: joerg.wegener@med.uni-goettingen.de; slehnart@med.uni-goettingen.de

RyR channel pore-blocking drug approved for treatment of malignant hyperthermia, exhibited antiarrhythmic effects in some patients with RyR2 mutations (Penttinen et al, 2015) and patients with heart failure (Hartmann et al, 2017) but may be impractical for chronic treatment (Bers, 2017) because direct dantrolene actions were questioned at the single-channel level (Choi et al, 2017). The water-soluble so-called RyCal S107 inhibited both RyR2-mediated $Ca^{2+}$ leak and stress-induced ventricular arrhythmias in RyR2$^{R2474S/+}$ mice (Lehnart et al, 2008) and reduced heart failure progression in homozygous RyR2$^{S2808D+/+}$ mice carrying a mutation that mimics PKA hyper-phosphorylation (Shan et al, 2010). Recently, the RyCal S36 corrected $Ca^{2+}$ leak in human iPSC-derived cardiomyocytes from a patient with CPVT (RyR2-E4076K) by reducing RyR2-mediated $Ca^{2+}$ spark activity and prohibited arrhythmia but not the progression of heart failure in mice. (Mohamed et al, 2018). Thus, at the moment, current therapeutic options for RyR2 modulation seem to be promising but may be contaminated with additional mechanisms of action (Connell et al, 2020).

In this study, we focused on the development and identification of a new class of multi-targeted compounds related to 1,4-benzothiazepine derivatives that combine activation of SR $Ca^{2+}$ uptake with reduction of $Ca^{2+}$ spark activity in cardiomyocytes the latter being shown to strongly correlate with $Ca^{2+}$ leak (Walker et al, 2014). Clinical studies on both SERCA2a overexpression and treatment with the SERCA2a activator istaroxime have been found to be safe in heart-failure patients (Jessup et al, 2011; Carubelli et al, 2020). In contrast to the latter compound that acts on SERCA2a and $Na^+$/$K^+$ ATPase (Micheletti et al, 2007), here we identify the first multi-targeted small compound with pronounced SR $Ca^{2+}$ leak inhibitory and SR $Ca^{2+}$ uptake stimulatory action in cardiomyocytes from mice and human, together with a prominent action in a human-associated mouse model of CPVT, thus opening avenues for the development of a novel class of RyR2 multi-target–directed ligands.

## Results

### Screening for new multi-targeted compounds

First, the effects of 17 newly synthetized derivatives together with the RyCal compound S36 (Fig 1A) on intracellular $Ca^{2+}$ signals were investigated in a plate reader assay using HEK293 cells that express both the RyR2 $Ca^{2+}$ release channel and the luminal calcium indicator R-CEPIA1er in the ER (Murayama et al, 2018). As expected, RyR2-expressing HEK293 cells showed reduced background-normalized $Ca^{2+}$ fluorescence signals in the ER ([$Ca^{2+}$]$_{ER}$) explained by the increased $Ca^{2+}$ leak through the exogenously expressed, spontaneously opening RyR2 channels; if a drug compound would inhibit this spontaneous RyR2 activity, the ER $Ca^{2+}$ signals monitored by the calcium indicator R-CEPIA1er will show an increased signal (Murayama et al, 2018). Accordingly, Fig 1B shows the effects of the new compounds together with S36, S107, and the proposed SERCA2a activator CDN1163 (Cornea et al, 2013) on the ER $Ca^{2+}$ signals in the induced RyR2-expressing HEK293 cells. Seven of the new compounds, and S107 and CDN1163, did not influence the ER $Ca^{2+}$ signals (grey bars). Notably, 10 of the new small chemical

compounds significantly increased ER $Ca^{2+}$ signals resulting in an S36-comparable or apparently greater effect (orange and green bars). Out of these, the new compound GM1869 (Fig 1A) significantly, concentration-dependently, and most consistently increased the ER $Ca^{2+}$ signal in this assay, indicating that this compound may work as potent inhibitor of spontaneous RyR2-dependent ER $Ca^{2+}$ leak (Fig 1C).

In a second approach, the effects of the inhibitory compounds from Fig 1 were investigated on SR $Ca^{2+}$ load in cardiac HL-1 cells (Min et al, 2012). In principle, HL-1 cells were loaded with the intracellular $Ca^{2+}$ indicator FLIPR (FLIPR Calcium 6 Assay Kit) and stimulated quickly with a high concentration of caffeine (10 mM) to empty the intracellular ER $Ca^{2+}$ store and, consequently, allow an indirect estimation of the SR $Ca^{2+}$ content (Smith et al, 1988; Reggiani, 2021); if a drug compound affects the SERCA2a $Ca^{2+}$ pump activity, the SR $Ca^{2+}$ content will be altered because of a change in the SR $Ca^{2+}$ filling, monitored by the magnitude of caffeine-induced $Ca^{2+}$ release. Notably, most of the derivatives listed in Fig 1B did not modify SR $Ca^{2+}$ load in this assay except GM1840 and GM1869 (Fig S1). However, because GM1840 was less efficient than GM1869 on $Ca^{2+}$ leak in RyR2-expressing HEK293 cells (see Fig 1B) and exhibited less water-soluble properties than GM1869, we focused on GM1869 in further experiments. Fig 2A shows the concentration-dependent effects of GM1869 through four orders of magnitude on the caffeine-induced $Ca^{2+}$ release in HL-1 cells. Indeed, HL-1 treatment with GM1869 significantly and concentration-dependently increased the amount of released $Ca^{2+}$ in HL-1 cells (Fig 2B) indicating that this compound enhanced SR $Ca^{2+}$ uptake via increased SERCA2 activity indicating a possibly unique multi-targeted activity of GM1869 on both SR $Ca^{2+}$ uptake in HL-1 and ER $Ca^{2+}$ leak in HEK293 cells, respectively.

### GM1869 significantly decreased RyR2-mediated $Ca^{2+}$ spark activity in permeabilized mouse cardiomyocytes

Next, we aimed for a more physiological target validation of the selected 1,4-benzothiazepine derivative GM1869 using isolated adult murine ventricular cardiomyocytes. Thus, we compared the effects of the known RyR2 pore blocker dantrolene, the RyR2-closed state-stabilizing S36 and the effects of the new derivative GM1869 on RyR2-mediated $Ca^{2+}$ spark activity in permeabilized living cardiomyocytes to ascertain rapid compound access to the intracellular targets and to control RyR2 channel gating by clamped cytosolic [$Ca^{2+}$] concentration according to Galimberti & Knollmann (2011). Fig 3 shows that both dantrolene (10 µM, Fig 3A) and S36 (10 µM, Fig 3B) significantly decreased the $Ca^{2+}$ spark frequency. Importantly, the novel derivative GM1869 (10 µM) also significantly decreased the $Ca^{2+}$ spark frequency in cardiomyocytes (Fig 3C) confirming the observed inhibition of spontaneous $Ca^{2+}$ leak in RyR2-expressing HEK293 cells (Fig 1C). Further analysis of the $Ca^{2+}$ spark shape revealed a reduced full-width-at-half maximum (FWHM) each for GM1869, dantrolene, and S36 (Fig S2A–C). Moreover, dantrolene but not S36 reduced the full-duration-at-half maximum (FDHM) also, whereas GM1869 slightly increased FDHM and sparked full duration (Fig S2D–I). Finally, under the given conditions, S107 was ineffective on $Ca^{2+}$ spark activity (Fig S3A and B), consistent with Fig 1B. Together, because RyR2-mediated $Ca^{2+}$ spark activity strongly

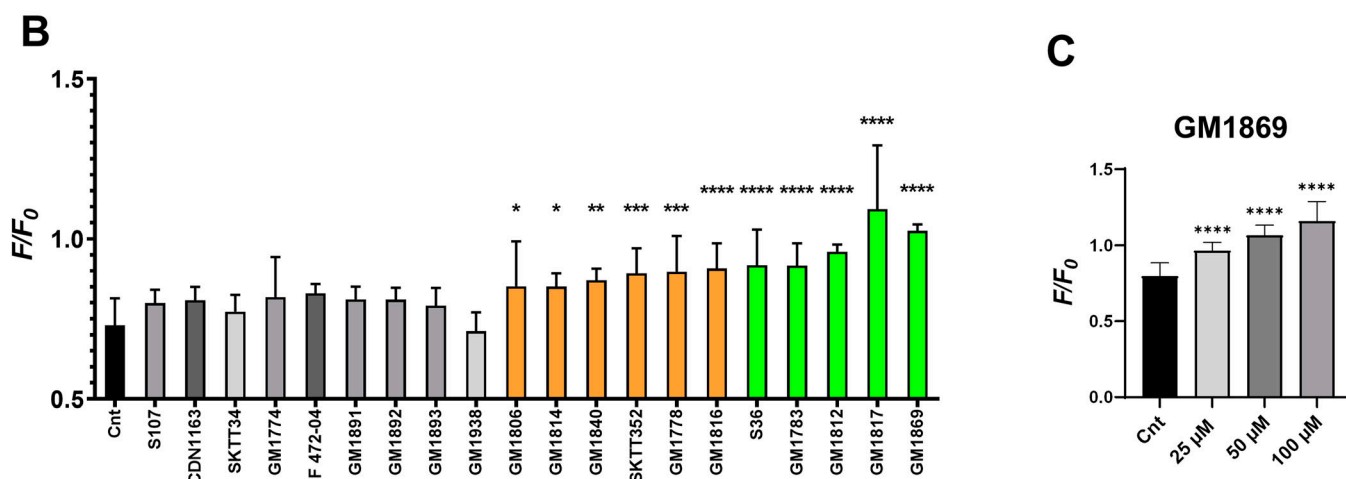

**Figure 1. 1,4-benzothiazepines reduce spontaneous Ca²⁺ leak in RyR2-expressing R-CEPIA1er HEK293 cells.**
**(A)** Chemical structures of S36 (patents US8710045B2 and EP1928850A2, [Marks et al, 2011]) and GM1869. **(B)** Fluorescence ($F/F_0$) in RyR2-expressing R-CEPIA1er HEK293 cells after the indicated compound (25 μM each, except 12.5 μM for GM1893) or vehicle (0.1% DMSO) application. $F/F_0$ is the ratio between the average fluorescence before (first 90 s) and after (last 100 s) injection of the test compound. Chemical structures of the compounds are presented in the supplementary information. No significant change in $F/F_0$ was observed with S107, CDN1163, SKTT34, GM1774, CF47204, GM1891, GM1892, GM1893, GM1938, and GM1806, (grey bars). Significant changes in $F/F_0$ indicating a reduction in spontaneous Ca²⁺ leak were observed by the derivatives GM1806, GM1814, GM1840, SKTT352, GN1778, and GM1816 (orange bars), and by the lead compound S36, GM1783, GM1812, GM1817, and GM1869 (green bars). **(C)** Concentration-dependent reduction of Ca²⁺ leak by GM1869 in RyR2-expressing R-CEPIA1er HEK293 cells. ER fluorescence ($F/F_0$) was significantly increased by increasing concentrations of GM1869 (non-paired experimental design). Values represent the mean ± SD with n = 17. Asterisks indicate statistical significant differences with ****$P < 0.0001$, **$P < 0.005$, *$P < 0.05$ versus vehicle (Cnt) by ordinary one-way ANOVA test.

correlates with SR Ca²⁺ leak (Walker et al, 2014), these data establish a RyR2 Ca²⁺ leak-inhibitory efficacy of the new compound GM1869.

## Compounds modulating SERCA2a-mediated Ca²⁺ wave decay in permeabilized CM

Subsequently, we assessed the effects of the multi-targeted compound on SERCA2a-mediated SR Ca²⁺ uptake. At a nominal Ca²⁺ concentration of 80 nM, permeabilized cardiomyocytes exhibited spontaneous Ca²⁺ waves at a frequency of ~6–18/min under our experimental conditions. The process of SR Ca²⁺ uptake was comparatively analyzed using identical segments of the Ca²⁺ wave signal before and after drug application (paired experimental design). Segments (1–2 μm) for analysis were chosen to avoid interference with wave propagation. Here, SR Ca²⁺ uptake is represented as the decay of the Ca²⁺ wave from the half-maximal peak amplitude to baseline obtained by a fit with a mono-exponential function giving the time constant (τ). All fits described the mono-exponential decay ($r^2 > 0.93$) well, thereby characterizing the activity of SERCA2a Ca²⁺ pumps and indicating no contamination by other processes (e.g., mitochondrial Ca²⁺ uptake). Fig 4 shows the effects of dantrolene, S36, and GM1869 (each at 10 μM) on the kinetics of Ca²⁺ uptake in permeabilized cardiomyocytes. Dantrolene significantly slowed down the

kinetics of Ca²⁺ uptake showing that this drug may inhibit SERCA2a activity (Fig 4A). S36 did not change the rate of Ca²⁺ uptake (Fig 4B). In contrast, GM1869 significantly accelerated the kinetics of Ca²⁺ uptake indicating that this drug promotes SERCA2a activity (Fig 4C). This view is further supported by our observation that Ca²⁺ wave velocity was significantly slowed down by dantrolene, not affected by S36, but significantly accelerated by GM1869 (Fig S4A–C and E–G). Indeed, a slower Ca²⁺ wave propagation has been shown to correspond to less SERCA2a activity in cardiomyocytes from SERCA2a knock-down mouse hearts (Stokke et al, 2010). Unexpectedly, the RyR2 inhibitor S107 significantly slowed SR Ca²⁺ uptake and Ca²⁺ wave velocity similar to dantrolene (Figs S3C and S4D and H), complementing our assessment of drug compounds that may inhibit SERCA2a activity.

## Drug effects in permeabilized CM from RyR2^R2474S/+ mice with increased SR Ca²⁺ leak

The heterozygous mutation in the *RYR2* gene leading to the amino acid exchange R2474S has been associated with the phenotype of CPVT type 1 (Priori et al, 2002). The corresponding knock-in mouse model RyR2^R2474S/+ has been shown to not only to recapitulate the human phenotype of exercise-induced ventricular tachycardias,

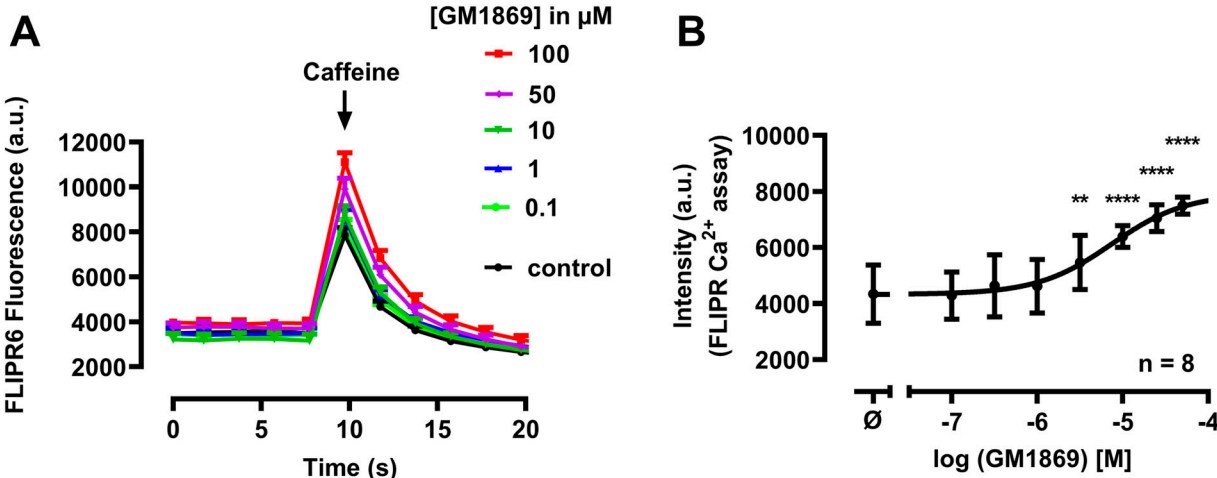

**Figure 2. The S36 derivative GM1869 enhances concentration-dependently caffeine-induced Ca$^{2+}$ release in HL-1 cells.**
**(A)** Original traces of changes in intensities monitored by the FLIPR Calcium6 assay using HL-1 cells after caffeine application in control conditions (DMSO, 0.1%) and in the presence of GM1869 at the indicated concentrations. Caffeine (10 mM) was applied at t = 10 s. **(B)** Maximal amplitudes of the Caffeine induced signal using the FLIPR Ca$^{2+}$ assay in HL-1 cells in control conditions and in the presence of GM1869 at the indicated concentrations. ∅ indicates the use of vehicle only. Data were fitted by a sigmoidal concentration response curve using GraphPad Prism. EC$_{50}$ value was calculated to 7.4 μM. Values represent the mean ± SD with n = 8. Asterisks indicate statistical significant differences with ****$P < 0.0001$, **$P < 0.001$ versus DMSO (Cnt) by ordinary one-way ANOVA test.

but to exhibit a profoundly increased SR Ca$^{2+}$ leak during increased catecholaminergic stress (Lehnart et al, 2008; Shan et al, 2012). In our experimental conditions, the Ca$^{2+}$ spark frequency, FWHM, and the Ca$^{2+}$ wave velocity were not significantly different between WT and RyR2$^{R2474S/+}$ cardiomyocytes (Fig S5A–C) whereas the Ca$^{2+}$ spark FDHM was significantly increased in RyR2$^{R2474S/+}$ cardiomyocytes (Fig S5D). Therefore, we compared the effects of dantrolene, S36, and GM1896 on the RyR2-dependent Ca$^{2+}$ spark activity (corresponding to RyR2-mediated SR Ca$^{2+}$ leak [Walker et al, 2014]) and the SERCA2a-mediated kinetics of SR Ca$^{2+}$ uptake in permeabilized cardiomyocytes from WT and RyR2$^{R2474S/+}$ mice. Fig 5A–C summarize the effects of dantrolene, S36, and GM1869 (10 μM each) on the Ca$^{2+}$ spark frequency (in % of control) and Fig 5D–F summarize their effects on the kinetics of SR Ca$^{2+}$ uptake (relative changes to control). Each in WT and RyR2$^{R2474S/+}$ cardiomyocytes, dantrolene significantly reduced the Ca$^{2+}$ spark frequency to a similar degree (Fig 5A), whereas it delayed likewise the kinetics of SR Ca$^{2+}$ uptake significantly (Fig 5D). Interestingly, both S36 and GM1869 significantly decreased the Ca$^{2+}$ spark frequency in WT CMs but acted significantly stronger in RyR2$^{R2474S/+}$ CMs (Fig 5B and C). Whereas S36 had no significant effect on the kinetics of SR Ca$^{2+}$ uptake (Fig 5E), GM1869 accelerated the kinetics of SR Ca$^{2+}$ uptake significantly likewise in WT and RyR2$^{R2474S/+}$ CMs (Fig 5F). These data indicate that GM1869 exhibits more pronounced dual-acting properties in RyR2$^{R2474S/+}$ CMs, suggesting that it may represent a new lead pharmacological substance for a multi-targeted therapy of CPVT type 1.

### S36 and GM1869 modulate electrically evoked Ca$^{2+}$ transients in intact CMs

In the next series of experiments, we comparatively studied the effects of S36 and GM1869 on electrically evoked Ca$^{2+}$ transients in intact adult murine ventricular CMs to get insights in the pharmacokinetic properties of the drugs. For this purpose, cells were loaded with the intracellular Ca$^{2+}$ indicator Fluo-4 AM (5 μM) and paced five times at 1 Hz every 2 min after drug application. Ca$^{2+}$ transients were analyzed by mono-exponential functions, each with respect to the rise time (from baseline to peak) mainly reflecting RyR2-mediated Ca$^{2+}$ release and the decay time (from half-maximal peak to baseline) mainly reflecting SERCA2a activity. Fig 6 shows representative Ca$^{2+}$ transients each for the control and S36 (Fig 6A) and GM1869 (Fig 6B) treated conditions (10 μM each). Analysis of the Ca$^{2+}$ transient rise and decay kinetics showed that after 6 min of S36 treatment significantly slowed down the Ca$^{2+}$ release, whereas Ca$^{2+}$ decay was not influenced (Fig 6A). Interestingly, 6-min treatment with GM1869 significantly delayed the rise time but also significantly accelerated the decay time of the Ca$^{2+}$ transients indicating that GM1869 affects both RyR2 and SERCA2a functions in intact cardiomyocytes (Fig 6B). Notably, 6-min drug compound exposure needed for the development of the drug effects seems to be fairly acceptable for a multi-targeted pharmacokinetic process of GM1869, which depends on membrane permeation and intracellular binding to its putative targets, namely SERCA2a and RyR2.

### S36 and GM1869 modulate Ca$^{2+}$ transients in human iPSC-CMs

To translate the multi-targeted compound effects to a human model system, we investigated the effects of S36 and GM1869 on electrically evoked Ca$^{2+}$ transients in human iPSC-derived cardiomyocytes (iPSC-CMs) matured for at least 60 d. Again, Ca$^{2+}$ transients were analyzed by mono-exponential functions to characterize the intracellular Ca$^{2+}$ release and decay kinetics. Fig 7 shows exemplary Ca$^{2+}$ transients of untreated control and S36

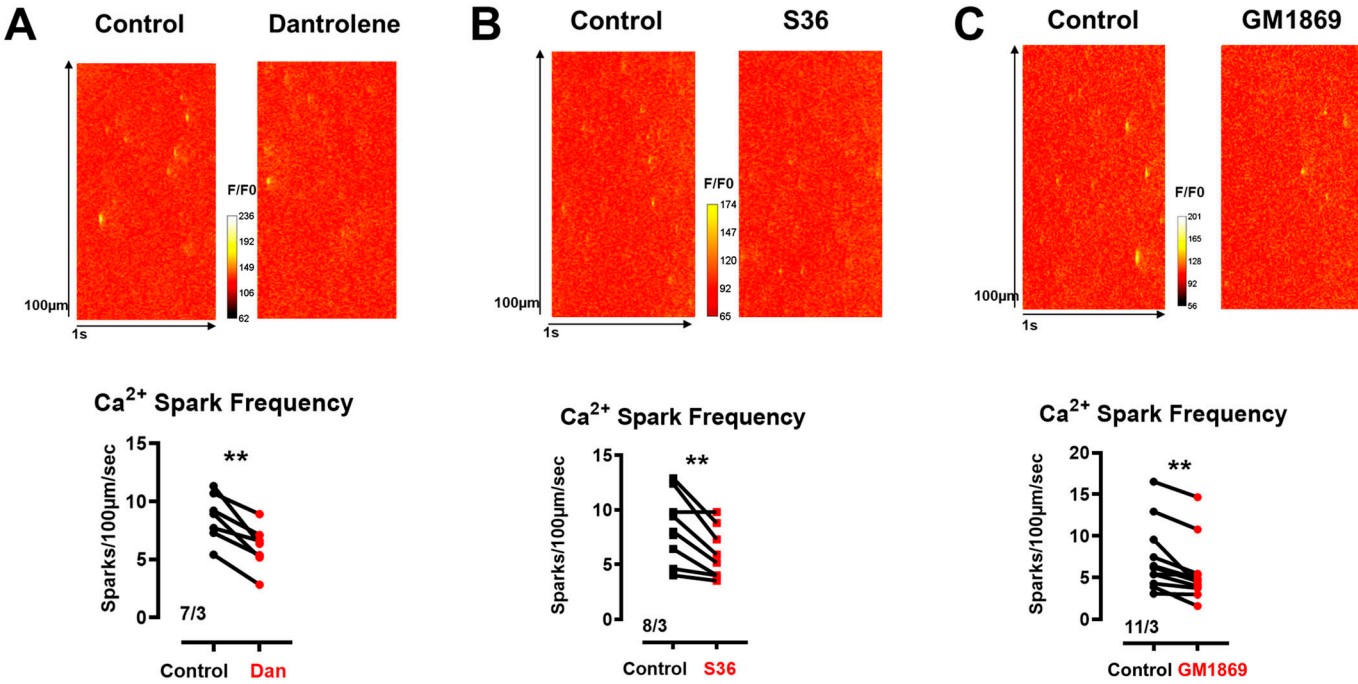

**Figure 3. Dantrolene (Dan), S36, and GM1869 reduced Ca²⁺ spark activity in saponin-permeabilized ventricular cardiomyocytes (sp-vCM) from WT mice.**
(A, B, C, top) Representative confocal line scans of Ca²⁺ spark activity in the absence (Ctr) and presence of 10 µM Dan (A), 10 µM S36 (B), and 10 µM GM1869 (C) in sp-vCM from WT mice. (A, B, C, bottom) Ca²⁺ spark frequencies in the absence (Ctr) and presence of 10 µM Dan (A), 10 µM S36 (B), and 10 µM GM1869 (C) in sp-vCM from WT mice. Vertical lines connect the respective data pairs. Numbers correspond to the number of cells/number of mice. Datasets were analyzed by paired $t$ test. **$P < 0.01$ with $P = 0.002$ for Dan, $P = 0.008$ for S36, and $P = 0.002$ for GM1869.

(Fig 7A) or GM1869 (Fig 7B) treated hiPSC-CMs. The rise and decay kinetics were analyzed 4 min each after S36 and GM1869 application (0.1 µM each, Fig 7C–F). Similar to intact murine cardiomyocytes, application of S36 did not change the decay time but significantly slowed down the Ca²⁺ release kinetics of the Ca²⁺ transients. In contrast, GM1869 significantly both delayed the rise time and accelerated the decay time of the Ca²⁺ transients indicating that GM1869 affects both the RyR2 and SERCA2a functions also in human iPSC-CMs.

To get more insights towards the affinity of our compound, we performed concentration–response curves for S36 and GM1869 in RyR2-WT and RyR2-R2474S⁺/⁻ iPSC-derived cardiomyocytes (Fig 7G–J). Ca²⁺ transients in iPSC-CM were recorded in control conditions and in the presence of three cumulatively applied drug concentrations after 2 min each for S36 and GM1869, respectively. Rise times and decay times obtained in each cell were normalized to the values obtained under control conditions. The fit of the datasets with concentration–response curves estimated the IC₅₀ values for the rise times to 150 pM (WT) and 40 pM (R2474S⁺/⁻) for S36 (Fig 7G) and to 25 pM (WT) and 45 pM (R2474S⁺/⁻) for GM1869 (Fig 7I). EC₅₀ values for the decay times were estimated to 97 pM (WT) and 71 pM (R2474S⁺/⁻) for GM1869 (Fig 7J), whereas no EC₅₀ values were obtained for the datasets with S36 (Fig 7H). These data indicate that the affinity of GM1869 may be sufficient for clinical application as an orally active drug.

## Discussion

The present study identified a new multi-targeted small chemical compound (GM1869) that acts simultaneously by both decreasing SR Ca²⁺ release via modulating systolic RyR2 channel function and by stimulating SR Ca²⁺ uptake via SERCA2 pump activity in murine cardiomyocytes and in human iPSC-CMs. Strikingly, this GM1869 substance exhibited its most pronounced effect on RyR2 activity in cardiomyocytes isolated from hearts of an in-depth characterized CPVT-susceptible RyR2^R2474S/⁺ knockin mouse model (Lehnart et al, 2008) compared with WT littermate cardiomyocytes. Thus, we suggest that GM1869 is a novel lead candidate for developing multi-targeted drug ligands as proposed previously (Zhang et al, 2023) that may allow further fine-tuning of their effects either towards RyR2 Ca²⁺ leak or SERCA2a pump activity, respectively, dependent on the pathophysiological context, for example, heart failure affecting ~2% of the general population with its known hallmarks of RyR2 channel and SERCA2a pump dysfunction (Braunwald, 2015).

Dysregulation of intracellular Ca²⁺ cycling, particularly SR Ca²⁺ leak, is a phenomenon observed in a number of heart disease phenotypes (Dridi et al, 2020; Hamilton et al, 2021). Increased catecholamine-induced acute SR Ca²⁺ leak has been extensively discussed for CPVT type 1 in which diastolic Ca²⁺ leak occurs via abnormally enhanced activity of mutant tetrameric RyR2 channels, promoting delayed afterdepolarizations, atrial and ventricular arrhythmias, and sudden cardiac death (Wleklinski et al, 2020). In addition, in a canine model of heart failure progression, an early

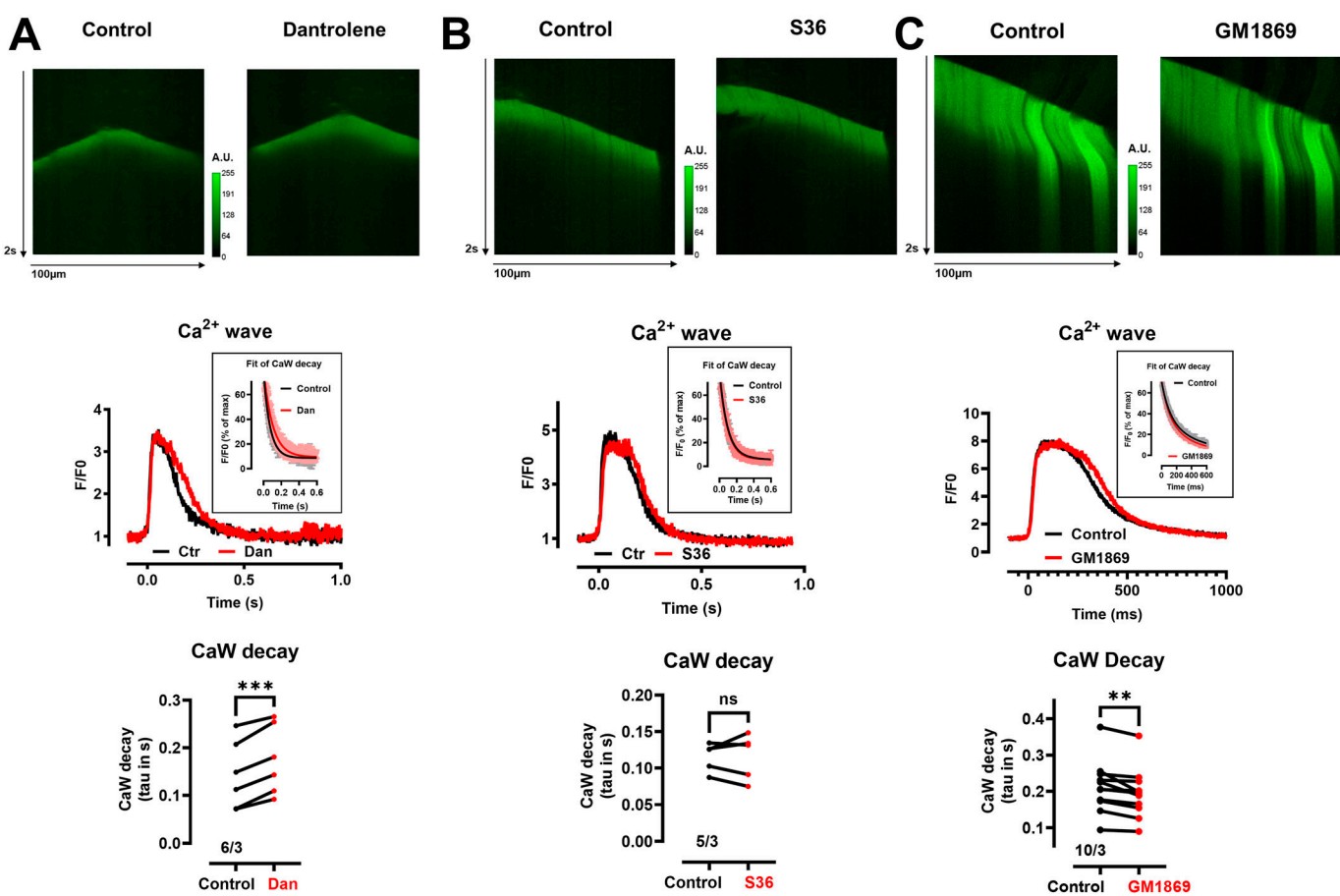

**Figure 4. Dantrolene (Dan) and GM1869, but not S36, modulate Ca²⁺ uptake in saponin-permeabilized ventricular cardiomyocytes (sp-vCM) from WT mice.**
(A, B, C, top) Representative confocal line scans of spontaneous Ca²⁺ waves in the absence (Control) and presence of 10 μM Dan (A) 10 μM S36 (B), and 10 μM GM1869 (C) in sp-vCM from WT mice. (A, B, C, middle) Averaged Ca²⁺ transients from spontaneous Ca²⁺ waves in the absence (Control) and presence of 10 μM Dan (A) 10 μM S36 (B), and 10 μM GM1869 (C) in sp-vCM from WT mice. The obtained Ca²⁺ transients were averaged from three to four independent experiments and shown as indicated. Single Ca²⁺ transients were obtained by averaging the identical 1–2 μm segments of the Ca²⁺ wave in the absence and presence of the drug to avoid the influence of wave propagation and to reduce the noise in the signal. Insets show the respective mono-exponential fits to the mean data ± SEM to determine the overall decay times. (A, B, C, bottom) Decay times of Ca²⁺ transients in the absence (Ctr) and presence of 10 μM Dan (A), 10 μM S36 (B), and 10 μM GM1869 (C) in sp-vCM from WT mice. Decay times correspond to the tau values (in s) obtained after a mono-exponential fit of the decay of the signal from half maximum to baseline. Vertical lines connect the respective data pairs. Numbers correspond to the number of cells/number of mice. Datasets were analyzed by paired $t$ test. ***$P < 0.001$; **$P < 0.01$; n.s., nonsignificant; with $P = 0.0008$ for Dan, $P = 0.94$ for S36, and $P = 0.004$ for GM1869.

and continuous increase in RyR2-mediated SR Ca²⁺ leak has been proposed (Belevych et al, 2011), although this defect has been mainly assigned to decreased coupling of Ca²⁺ influx and RyR2-mediated Ca²⁺ release evidenced by a loss of canonical dyadic nanodomain organization (Manfra et al, 2017) and increased junctophilin-2 cleavage by calpain (Lehnart & Wehrens, 2022; Weninger et al, 2022). Finally, increased RyR2-mediated Ca²⁺ leak has been associated with ageing and cardiac dysfunction involving enhanced RyR2 glycation (Ruiz-Meana et al, 2019).

Pharmacological treatment of pathologically increased SR Ca²⁺ leak is currently indirect through β-adrenergic receptor blockers as first-line, which can be seconded by anti-arrhythmic drugs like flecainide or propafenone as outlined by consensus guidelines for CPVT patients (Priori et al, 2013; Baltogiannis et al, 2019). Interestingly, both flecainidine and propafenone, but not lidocaine, have been shown to inhibit RyR2 activity besides their effects on sodium channels (Hwang et al, 2011). However, it is

controversial whether the flecainide effect in CPVT is because its action on the RyR2 channel pore or whether sodium channel block alone is responsible for the beneficial effect (Bannister et al, 2015). It has been argued that flecainide's inhibitory potency on RyR2 activity in permeabilized CMs or on single RyR2 channels might be too low in intact cells to explain its clinical efficacy, and that sodium channel blockers lacking effects on RyR2 activity, such as tetrodotoxin, may decrease spontaneous SR Ca²⁺ leak via increased NCX Ca²⁺ extrusion (Sikkel et al, 2013). Thus, although both anti-arrhythmic drugs are effective clinically, their use as potential first-line therapeutics has still to be clarified towards potential additive effects on sodium channel versus RyR2 modulation (Watanabe et al, 2009; Behere & Weindling, 2016).

Therefore, it is timely to develop and identify allosteric pharmacological RyR2 inhibitory compounds as potential first-line and mechanism-directed therapeutically active drugs.

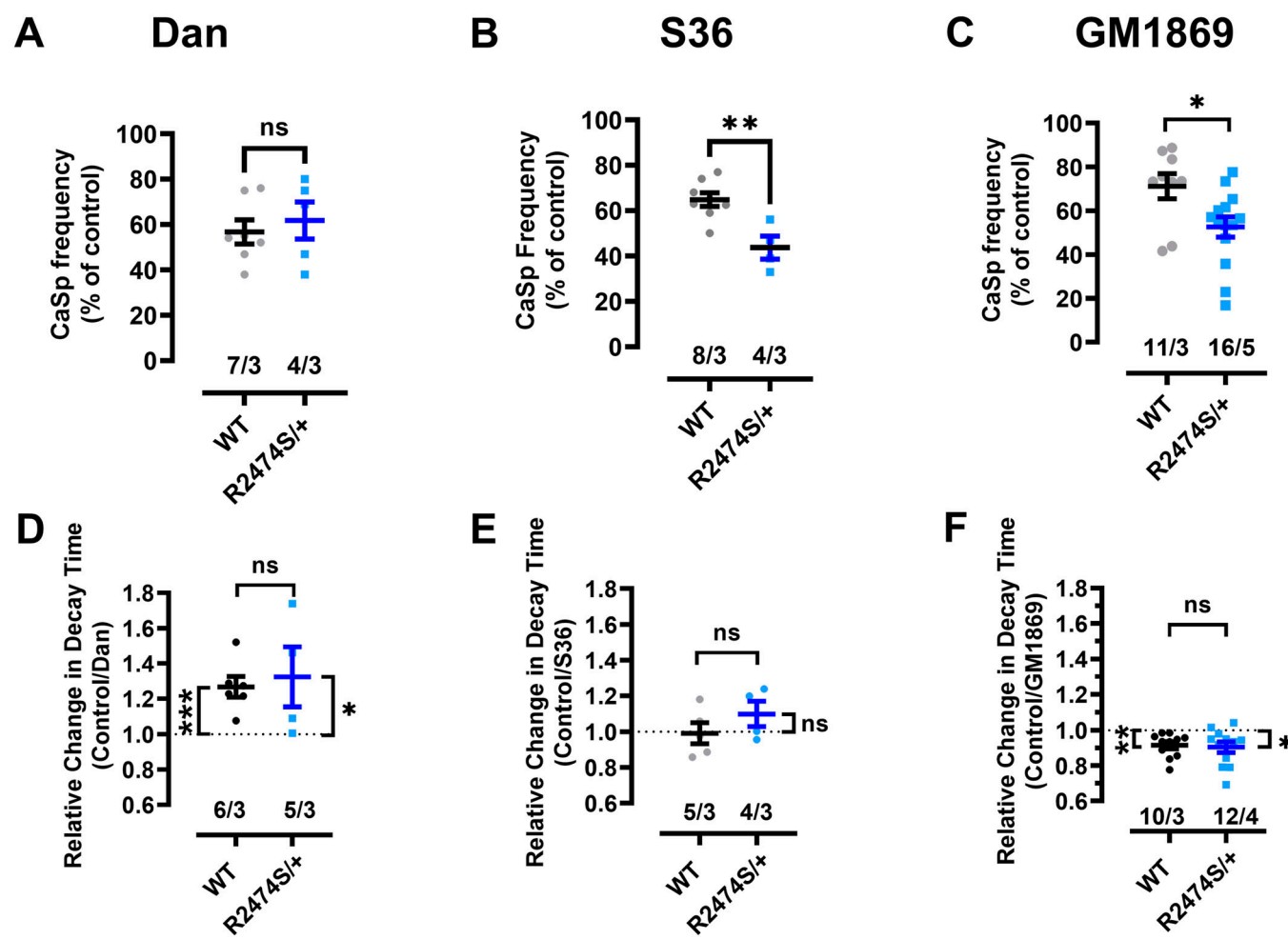

**Figure 5. Dantrolene (Dan), S36, and GM1869 modulate differentially Ca²⁺ spark activity and Ca²⁺ uptake in saponin-permeabilized ventricular cardiomyocytes (sp-vCM) from WT and RyR2^R2474S/+ mice.**

**(A, B, C)** Reduction of Ca²⁺ spark frequency (in % of control) by Dan ((A), 10 μM), S36 ((B), 10 μM), and GM1869 ((C), 10 μM) in cardiomyocytes from WT and RyR2^R2474S/+ mice. Numbers correspond to the number of cells/number of mice. Please note that the data shown for WT correspond to the respective data pairs from Fig 3 (Ca²⁺ spark activity). Datasets were analyzed by unpaired *t* test. n.s., nonsignificant with *P* = 0.59 for Dan, **\**P* < 0.01 with *P* = 0.003 for S36, and \**P* < 0.05 with *P* = 0.019 for GM1869. **(D, E, F)** Relative changes in the decay time tau of Ca²⁺ transients from spontaneous Ca²⁺ waves (in control/drug) by Dan ((D), 10 μM), S36 ((E), 10 μM), and GM1869 ((F), 10 μM) in cardiomyocytes from WT and RyR2^R2474S/+ mice. Data points for Dan (D) were significantly above 1 for both mice strains (paired *t* test with *P* = 0.0008 for WT and *P* = 0.03 for RyR2^R2474S/+ mice). Data points for S36 (E) were not significantly different from 1 in cardiomyocytes from WT and RyR2^R2474S/+ mice (paired *t* test with *P* = 0.94 for WT and *P* = 0.31 for RyR2^R2474S/+ mice). Data points for GM1869 (F) were significantly below 1 in cardiomyocytes from WT and RyR2^R2474S/+ mice (paired *t* test with *P* = 0.004 for WT and *P* = 0.02 for RyR2^R2474S/+ mice). Numbers correspond to the number of cells/number of mice. Please note that the data shown for WT correspond to the respective data pairs from Fig 4 (Ca²⁺ decay times). Datasets between mouse strains were analyzed by unpaired *t* test. n.s., nonsignificant with *P* = 0.72 for Dan, with *P* = 0.28 for S36, and *P* = 0.77 for GM1869.

Under physiological conditions, RyR2 opening is induced by the mechanism of Ca²⁺-induced Ca²⁺ release (Meissner, 2017). The sensitivity of RyR2 to intracellular [Ca²⁺] is tightly regulated by protein–protein interactions including FKBP12.6 (a.k.a. calstabin-2), calmodulin (CaM), and calsequestrin (CASQ2). At a systemic level of regulation, the association of FKBP12.6 with the tetrameric macromolecular RyR2 channel complex is especially thought to serve as a potential therapeutic strategy against heart failure (Yano et al, 2003) because enhanced binding reduced, whereas diminished binding increased RyR2-mediated Ca²⁺ leak, the latter being affected by PKA-mediated phosphorylation of RyR2 during β-adrenergic stimulation (Marx et al, 2000; Marx & Marks, 2013). A weakened association between

FKBP12.6 and mutated RyR2 was also proposed to be related to CPVT type 1 pathology (Lehnart et al, 2008; Shan et al, 2012). Vice versa, small chemical molecules that directly enhance the association of FKBP12.6 to RyR2, namely JTV-519 (K201) and S107, improved cardiac muscle function in experimental models of heart disease (Wehrens et al, 2005; Shan et al, 2010). However, S107, one of the less effective drug compounds (US8710045B2), did not change RyR2-mediated Ca²⁺ spark activity in this study using permeabilized WT cardiomyocytes. In addition, no differences in FKBP12.6 binding were detected in recent cryo-EM structures for WT versus RyR2–R2474S channels, open or closed states or PKA-phosphorylated versus dephosphorylated channels (Miotto et al, 2022). Thus, despite promising early results

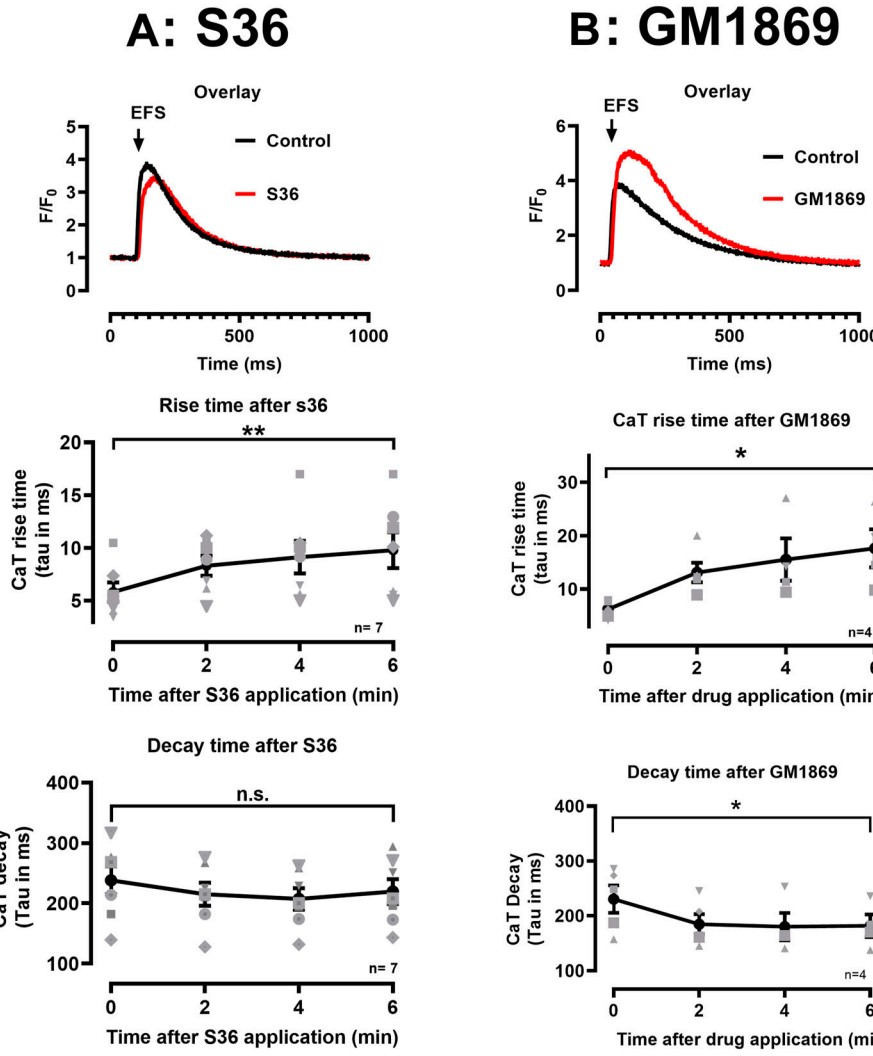

**Figure 6. S36 and GM1869 modulate differentially electrically induced Ca²⁺ transients in intact adult cardiomyocytes.**

(A, B, top) Overlay of representative Ca$^{2+}$ transients elicited by electrical field stimulation in control conditions and in the presence of S36 ((A), 10 μM) and GM1869 ((B), 10 μM) in cardiomyocytes from WT mice. (A, B, middle) Time-dependent changes in rise times of Ca$^{2+}$ transients in the presence of S36 (A) and GM1869 (B). Rise times represent the tau values calculated by a mono-exponential fit of the Ca$^{2+}$ transients from start to the peak of the signal. Ca$^{2+}$ transients were recorded in control conditions (time point 0) and in 2-min time intervals after drug application. $**P = 0.009$; $*$, $P = 0.04$, evaluated by paired $t$ test. (A, B, bottom) Time-dependent changes in decay times of Ca$^{2+}$ transients in the presence of S36 (A) and GM1869 (B). Decay times represent the tau values calculated by a mono-exponential fit of the Ca$^{2+}$ transients from half maximum to baseline of the signal. Ca$^{2+}$ transients were recorded in control conditions (time point 0) and in 2-min time intervals after drug application. Numbers correspond to the number of cells/number of mice. Datasets were analyzed by paired $t$ test. n.s., $P = 0.16$; $*P = 0.04$.

with K201 and subsequently S107, it is not clear to which extent an intervention leading to enhanced FKBP12.6 binding to RyR2 may contribute to an overall improvement of heart disease, particularly in larger clinically relevant species (Seidler et al, 2011) or in human heart models studied here. Because our drug screening method used HEK293 cells that do not endogenously express FKBP12.6 (Xiao et al, 2007), we assume that the described effects of GM1869 do not depend on FKBP12.6–RyR2 interactions.

Interestingly, we found that both S36 and GM1869 displayed a significantly more pronounced inhibitory action on RyR2-mediated Ca$^{2+}$ spark activity in murine RyR2$^{R2474S/+}$ compared with murine WT cardiomyocytes indicating a potential treatment benefit for CPVT type 1 (Wleklinski et al, 2020). However, we could not observe these clear-cut differences in the rise times of Ca$^{2+}$ transients from RyR2$^{R2474S/+}$ and WT iPSC-CM probably because of the different read out of RyR2 activity (Ca$^{2+}$ spark activity vs. rise times of Ca$^{2+}$ transients) and the immature status of iPSC-CM (Jiang et al, 2018).

Recently, a second generation 1,4-benzothiazepine derivative (ARM210) was shown to prevent fatal cardiac arrhythmias and to reduce SR Ca$^{2+}$ leak in RyR2$^{R2474S/+}$ mice (Miotto et al, 2022). In cryo-EM studies, this compound reversed inappropriate openings of RyR2–R2474S channels by stabilizing the closed conformation (Miotto et al, 2022). Furthermore, another cryo-EM study revealed binding of ARM210 to the cleft of the RY1&2 domain of RyR1, attenuated by engineered point mutations of RY1&2, indicating that ARM210 binds to a specific site with is homologous in RyR2 (Melville et al, 2022). The RY1&2 domain was postulated to represent a second ATP binding site in addition to the C-terminal ATP-binding site binding site described at the mutual junction of the S6c (cytoplasmic extension of S6), the CTD (C-terminal domain), and TaF (thumb and forefingers domain) domain (des Georges et al, 2016). In detail, ARM210 reversed the destabilization of the bridging solenoid domain (Hadiatullah et al, 2022) of RyR2 induced by the CPVT-associated RyR2–R2474S mutation thereby stabilizing the closed channel conformation (Miotto et al, 2022). Overall, the mechanism of ARM210 binding to RY1&2 occurred cooperatively together with ATP and stabilized the closed state in the presence of activating ligands (Ca$^{2+}$, ATP, and caffeine) (Melville et al, 2022).

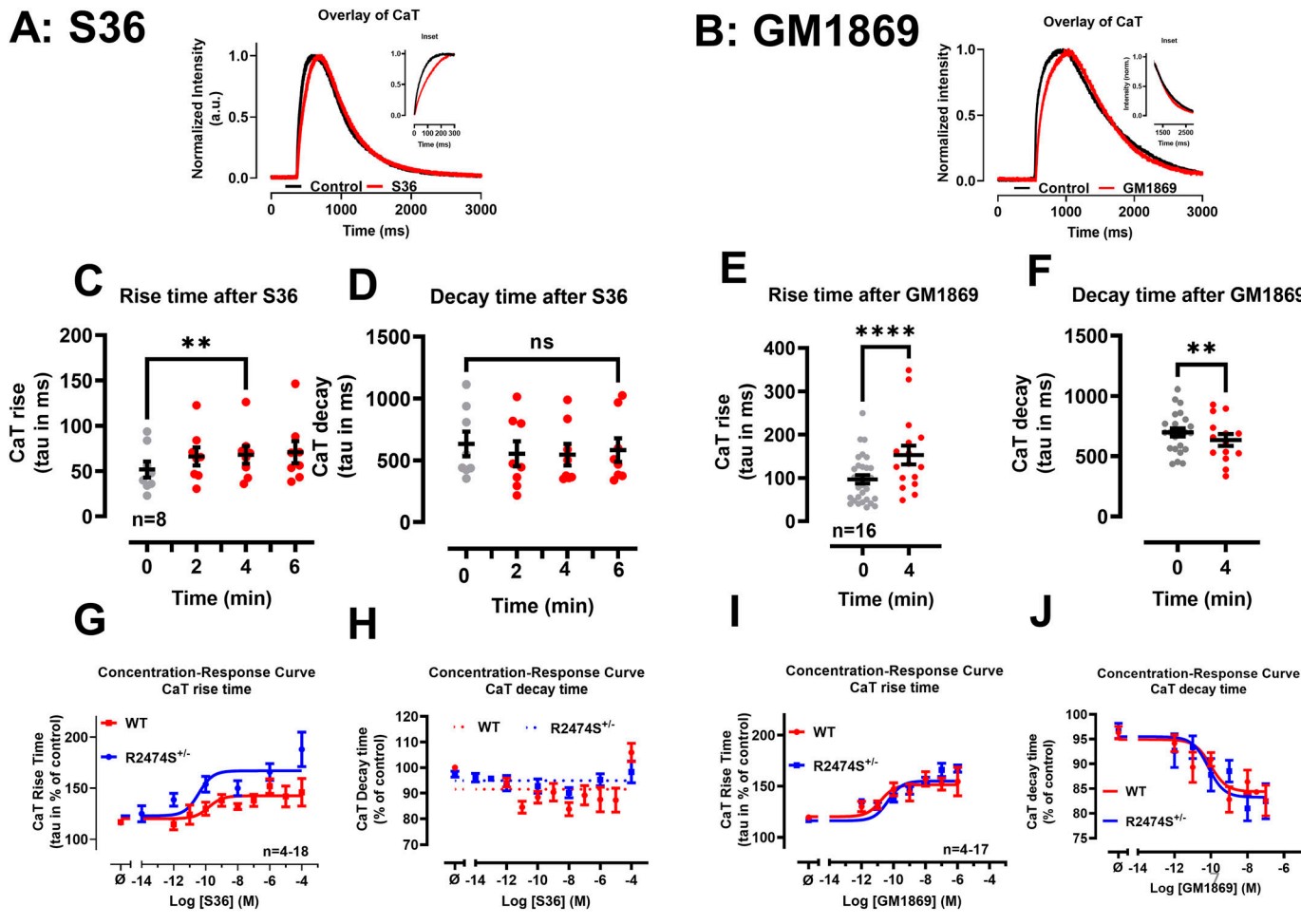

**Figure 7.  S36 and GM1869 modulate differentially Ca²⁺ transients in intact human iPSC-derived cardiomyocytes.**
**(A, B)** Overlay of representative Ca²⁺ transients in control conditions and in the presence of S36 ((A), 0.1 μM) and GM1869 ((B), 0.1 μM) in iPSC-derived cardiomyocytes. **(A, B)** Insets show the overlay of the Ca²⁺ transients from start to peak (A) and from 70% of the maximum to baseline (B). **(C, E)** Changes in rise times of Ca²⁺ transients in the presence of S36 (C) and GM1869 (E). Rise times represent the tau values calculated by a mono-exponential fit of the Ca²⁺ transients from start to the peak of the signal. Ca²⁺ transients were recorded in control conditions and every 2 min (S36) or at 4 min (GM1869) after drug application. **P = 0.0065 for S36; ****P = 0.000019 for GM1869, evaluated by paired t test. **(D, F)** Changes in decay times of Ca²⁺ transients in the presence of S36 (D) and GM1869 (F). Decay times represent the tau values calculated by a mono-exponential fit of the Ca²⁺ transients from half maximum to baseline of the signal. Ca²⁺ transients were recorded in control conditions and every 2 min (S36) or at 4 min (GM1869) after drug application. Datasets were analyzed by paired t test. n.s., P = 0.14; **P = 0.0043. **(G, H, I, J)** Concentration-response curves for S36 (G, H) and GM1869 (I, J) on rise times (G, I) and decay times (H, J) of Ca²⁺ transients, respectively, in RyR2-WT (red) and RyR2-R2474S⁺/⁻ (blue) iPSC cardiomyocytes. Data from each experiment were normalized to the data from the first transient recorded. Each cell was treated cumulatively with three increasing drug concentrations. Data are shown as mean ± SEM. Lines represent a nonlinear fit of each data set using Graphpad Prism (with log[agonist] versus response [three parameters]). **(G, I)** IC₅₀ values for the rise times were estimated to 150 pM (WT) and 40 pM (R2474S⁺/⁻) for S36 (G) and to 25 pM (WT) and 45 pM (R2474S⁺/⁻) for GM1869 (I). **(J)** EC₅₀ values for the decay times were estimated to 97 pM (WT) and 71 pM (R2474S⁺/⁻) for GM1869 (J), whereas no EC₅₀ values were obtained for the data sets (WT and R2474S⁺/⁻) with S36. **(H)** Instead, datasets with S36 were manually approximated by the dotted lines shown (H). Numbers correspond to the number of cells investigated.

Because GM1869 used here is also a 1,4-benzothiazepine derivative, we anticipate a similar mechanism of action, for example, stabilizing the channel closed state. Indeed, both the observed decrease of Ca²⁺ spark FWHM and the pronounced amelioration of the Ca²⁺ spark frequency in RyR2^R2474S/+ CMs by GM1869 suggest a shift of RyR2 cluster ensembles towards a lower Ca²⁺ spark fidelity. However, this concept has to be clarified by future studies using mathematical homology modeling as described previously (Bender et al, 2020).

Besides chronically enhanced diastolic Ca²⁺ leak, heart failure is frequently accompanied by reduced SERCA2a function, promoting a decreased SR Ca²⁺ load vis-à-vis a propensity for diastolic

intracellular Ca²⁺ overload (Bers et al, 2006), the extent of which, however, may vary between distinct phenotypes (Hasenfuss et al, 1994; Hasenfuss, 1998). Although treatment of heart failure by positive inotropic drugs, such as β-adrenoreceptor agonists or PDE inhibitors, increased long-term mortality (Packer et al, 1991; O'Connor et al, 1999; Toma & Starling, 2010), directly targeting SERCA2a by overexpression gene therapy (Jessup et al, 2011) or by the small molecule activator istaroxime demonstrated absence of severe side-effects in patients with heart failure (Carubelli et al, 2020). Likewise, SUMO1 restitution by adeno-associated virus-mediated gene therapy maintained SERCA2a levels and significantly improved cardiac function in heart failure mice (Kho et al,

2011). Consequently, our strategy to identify multi-targeted drug compounds with RyR2 inhibitory activity combined with SERCA2a activatory activity is expected to represent a safe approach for treatment of heart failure. In addition, we think that this approach is attractive to reduce the risk that pharmacologically enhanced SR $Ca^{2+}$ uptake increases SR $Ca^{2+}$ load, and, as a consequence, augments the driving force for $Ca^{2+}$ leak via posttranslationally modified RyR2 channels in the diseased heart. Indeed the data show that GM1869 enhanced $Ca^{2+}$ uptake in human and murine cardiomyocytes, both from WT and RyR2$^{R2474S/+}$ mouse hearts and WT and RyR2$^{R2474S/+}$ human iPSC-CM. However, because the focus of this study has been a functional identification of a candidate compound after screening in transiently RyR2-expressing HEK293 cells, we cannot exclude that the positive effects of our compounds on SERCA2a $Ca^{2+}$ uptake may be contaminated by stimulatory actions on β-adrenergic signaling. Nevertheless, we did not observe different effects of S36 or dantrolene in the presence or absence of cyclic AMP in permeabilized cardiomyocytes, which renders this scenario unlikely. In addition, under similar conditions, it has been shown previously that increased cAMP does not influence SR $Ca^{2+}$ leak in permeabilized cardiomyocytes (Bovo et al, 2017). Yet, at the moment, the exact mode of action of GM1869 and related compounds on SERCA2a activity is unclear. Because the detailed crystal structures of SERCA2a and SERCA2b have been published (Sitsel et al, 2019; Zhang et al, 2021), further studies might identify whether GM1869 interacts with the ATP binding site of SERCA2a as likewise shown for the interaction of the RyCal ARM210 with the RyR2 (Miotto et al, 2022) or whether GM1869 relieved phospholamban inhibition from SERCA2a as proposed for istaroxime (Ferrandi et al, 2013).

Pathologically increased SR $Ca^{2+}$ leak through RyR2 dysfunction combined with decreased SERCA2a function represents hallmarks in acquired forms of cardiac diseases including heart failure (Braunwald, 2015; Hamilton et al, 2021). Hence, identification of the first multi-targeted compound GM1869 inhibiting RyR2-mediated $Ca^{2+}$ leak together with activating SERCA2a function may represent a critical breakthrough for pharmacological development. Future work and testing of GM1869-derived compound libraries will without doubt further improve this type of multi-targeted drugs in the context of rodent, pig, and human cardiomyocytes and tissue model systems and open avenues for translation towards first-in-man studies.

## Materials and Methods

### Animals

All procedures associated with animal care and treatment conformed to the institutional and governmental guidelines (Directive 2010/63/EU of the European Parliament) and were approved by local authorities (T11.2, veterinarian state authority [LAVES]). The mouse line used for the study carried the RyR2–R2474S knock-in mutation (Lehnart et al, 2008) and was back-crossed for at least 10 generations into the C57BL/6N background. Breeding of WT and heterozygous RyR2$^{R2474S/+}$ mice generated littermate male mice at 12–24 wk of age used for the experiments.

### Chemicals

Chemicals and reagents were purchased from Sigma-Aldrich unless otherwise stated. Rycal S36 was obtained from Endotherm (www.endotherm-lsm.com). The synthesis of the tested compounds is described in the Supplemental Data 1 (Marks et al, 2011; Elek et al, 2019).

### Ca$^{2+}$ leak measurements in RyR2 expressing R-CEPIA1*er* HEK293 cells

Measurements on $Ca^{2+}$ leak in R-CEPIA1*er* HEK293 cells expressing RyR2 after treatment with doxycycline (2 μg/ml, 24 h) were performed as described (Murayama & Kurebayashi, 2019). These cells were a generous gift from Takashi Murayama (Juntendo University School of Medicine, Tokyo) and cultured in black-walled, clear-bottom 96-well micro-plates (Corning) using Dulbecco's Modified Eagle medium covered with fibronectin (10 μg/ml; Roche) in a humidified incubator at 37°C and 5% $CO_2$ (50,000 cells per well). Changes in ER calcium concentrations $[Ca^{2+}]_{ER}$ were measured on a multi-well plate reader TECAN Spark 20 M at 37°C. The 1,000 × stock solutions of the test compounds were prepared in DMSO. After the culture medium was removed, 100 μl of Tyrode's solution containing 2 mM $CaCl_2$ was added to the cells. Fluorescence was evoked by 560 nm excitation wavelength and collected in a bottom-read mode at 610 nm. The time courses were recorded in each well. Data were recorded every 10 s, exposure: 20 flashes, excitation bandwidth: 20 nm, emission bandwidth: 20 nm. Test compounds or vehicle (DMSO, 0.1%) were applied at 90 s after start of the assay at the indicated concentrations. The ratio values of the averaged fluorescence intensities $F/F_0$ before and after drug application were taken for analysis.

### Caffeine-induced Ca$^{2+}$ release in HL-1 cells

HL-1 cells (SCC065; Merck KGaA, 50,000 per well) were cultured in black-walled, clear-bottom 96-well micro-plates (Corning) after coating with fibronectin according to the manufacturer's instructions. Cytosolic $Ca^{2+}$ concentrations were measured on a multi-well plate reader TECAN Spark 20 M using FLIPR Calcium 6 Assay Kit (Molecular Devices LLC) at 37°C. HL-1 cells were loaded with the FLIPR Calcium 6 dye for 2 h at 37°C, together with drugs or vehicle (DMSO, 0.1%) at the indicated concentrations. Fluorescence intensities (485 nm excitation, >525 nm emission) were continuously monitored in each well for 1 min. Caffeine (10 mM final concentration) was injected at 10 s. The differences in intensities between the caffeine-induced maximal $Ca^{2+}$ signal and baseline, $\Delta F_{Caff}$, were taken for analysis. The dose response curves were calculated using the software package GraphPad Prism version 8.3.1. All data are presented as mean ± SD from eight independent experiments. Y=Bottom + (Top-Bottom)/(1 + 10 ^ ((LogEC$_{50}$-X)*HillSlope)) was used where HillSlope describes the steepness of the curve, Top and Bottom are plateaus in the units of the Y axis.

### Handling of human iPSC-derived cardiomyocytes

The human iPSC line is WT1. Bld2 (GOEi014-A.2) was differentiated into iPSC-vCMs as described (Cyganek et al, 2018) and also used to

generate the RyR2-R2474S$^{+/-}$ iPSC cell line that was kindly provided by L Cyganek (Stem Cell Unit, Department of Cardiology and Pulmonology, UMG Göttingen). In brief, differentiation was initiated at 80–90% confluence in Geltrex-coated plates with a cardio differentiation medium composed of RPMI 1640 with Glutamax and HEPES (Thermo Fisher Scientific), 0.5 mg/ml human recombinant albumin, and 0.2 mg/ml L-ascorbic acid 2-phosphate and sequential treatment with 4 $\mu$M CHIR99021 (Merck Millipore) for 48 h and then 5 $\mu$M IWP2 (Merck Millipore) for 48 h. The medium was changed to RPMI 1640 with Glutamax, HEPES, and 2% B27 (Thermo Fisher Scientific) at day 8. Metabolic CM selection was performed using RPMI 1640 without glucose (Thermo Fisher Scientific), 0.5 mg/ml human recombinant albumin, 0.2 mg/ml L-ascorbic acid 2-phosphate, and 4 mM lactate (Sigma-Aldrich) for 5 d. Afterwards, iPSC-CMs were cultured at least to day 60 for further maturation and then used in the experiments. All experiments on cardiomyocytes were performed at room temperature.

### Handling of adult cardiomyocytes

Ventricular cardiomyocytes were isolated from mice as described previously (Louch et al, 2011). Isolated cardiomyocytes were used directly after isolation according to Berisha et al (2021) or permeabilized with saponin as described (Guo et al, 2006). Briefly, 200 $\mu$l of the solution containing isolated cardiomyocytes was placed on a laminin-coated glass cover slip (Cell Systems) and allowed to settle down. After 10 min, cells were exposed to a relaxing solution containing (in mM): 1,2-bis(o-aminophenoxy)ethane-N,N,N′,N′-tetraacetic acid (BAPTA) 0.1, HEPES 10, K-aspartate 120, MgCl$_2$ 5.5, and adenosine Tri-phosphate di-Na$^+$ (di-Na$^+$ ATP) 5, KH$_2$PO$_4$ 5, phosphocreatine-diNa$^+$ 5, phosphocreatine-diTris 10, and creatine phosphokinase 10 U/ml. pH was adjusted to 7.2 with NaOH. After adding 0.03 mM Ca$^{2+}$, the free [Ca$^{2+}$] was estimated to 80 nM (Maxchelator). After 2 min, the supernatant was replaced with the relaxing solution containing saponin (40 $\mu$g/ml) for 1 min. Cells were then washed three times with the relaxing solution. The solution was replaced by the relaxing solution containing 10 $\mu$M Fluo-4 pentapotassium salt (Thermo Fisher Scientific). About 5% of the remaining cardiomyocytes exhibited spontaneous contractions and Ca$^{2+}$ waves at a constant frequency for at least 5 min.

### Ca$^{2+}$ imaging

Cardiomyocytes were imaged using an LSM 880 inverted microscope equipped with a 63x oil immersion objective (Zeiss). Fluo-4 was excited at 488 nm with a krypton/argon laser. The fluorescent emission was collected through a long-pass filter (>515 nm). All images were acquired digitally in line-scan mode (1.85 or 2.5 ms per line-scan; pixel size 0.1 $\mu$m) with 6,000 lines/image. The scan line was placed in parallel to the longitudinal axis of brick-shaped cardiomyocytes with clear cross striations that were solid attached to the cover slip. Images were recorded only from Fluo-4 loaded permeabilized cells that show spontaneous contractions and corresponding Ca$^{2+}$ waves at frequencies below 4 waves/10 s. These cells were attributed to have experienced a mild permeabilization procedure, that is, the cell membrane was sufficiently permeabilized to allow the membrane-impermeable Ca$^{2+}$ indicator

to enter the cell but the intracellular membranes and contractile structures were not affected, as evidenced by Ca$^{2+}$ release from intracellular stores and cell shortening, respectively. All measurements were performed under control conditions and after 2 min incubation with relaxing solution containing either vehicle (DMSO, 0.1%) or the indicated drug concentration. Drugs were added cumulatively if appropriate. Intact cardiomyocytes were loaded with Fluo-4 AM (5 $\mu$M; Thermo Fisher Scientific) for 45 min at room temperature and imaged as described above. Electrical field stimulation was performed as described (Wegener et al, 2021).

### Analysis

Images were analyzed using ImageJ (http://rsb.info.nih.gov/nih-image) equipped with the SparkMaster plugin (Picht et al, 2007). The detection criteria for Ca$^{2+}$ sparks were set at 3.8, that is, the threshold for the detection of events was 3.8 times the SD of the background noise divided by the mean. Ca$^{2+}$ spark amplitude was normalized as F/F$_0$ (with F$_0$ being the initial fluorescence recorded under steady-state conditions and $\Delta$F = F − F$_0$), duration was expressed as full duration or taken from the FDHM, and width was expressed as full width or taken from FWHM.

Ca$^{2+}$ spark shape (i.e., FWHW and FDHM) is reported to be different between RyR$^{+/+}$ and RyR$^{R2474S/+}$ channel clusters, especially with the observation of ember sparks in cardiomyocytes from mice expressing the RyR$^{R2474S/+}$ mutation (Danielsen et al, 2018). Spark shape is related to intracellular Ca$^{2+}$ buffer capacity (Stern et al, 2013), that is, using EGTA at low concentration increases the width and length of calcium sparks compared with using EGTA at high concentration (Bovo et al, 2015). Likewise, the use of the faster calcium chelator BAPTA reduces the width and length of Ca$^{2+}$ sparks compared with EGTA (Bovo et al, 2015). To compare the effects of the drugs tested more accurately, we used the fast calcium chelator BAPTA at 0.1 mM giving a free [Ca$^{2+}$] of 80 nM. In this condition, the frequency of Ca$^{2+}$ sparks and FWHM was not different in permeabilized cardiomyocytes from RyR2$^{+/+}$ and RyR$^{R2474S/+}$ mice, whereas FDHM was slightly increased in cardiomyocytes from RyR$^{R2474S/+}$ mice (Fig S5). However, analysis of drug effects should not be hampered by the differences in FDHM.

Spontaneous Ca$^{2+}$ waves were analyzed by calculating the wave speed and the Ca$^{2+}$ wave decay reflecting Ca$^{2+}$ uptake by SERCA2a activity. Ca$^{2+}$ wave speed was obtained by a linear approximation of wave propagation (in $\mu$m/ms). Ca$^{2+}$ decay was obtained in a 1–2 $\mu$m part of the wave (10–20 pixels in width) before and after drug application to exclude the influence of wave propagation but to reduce the noise in the recording. Because of permeabilization procedure, activity of sodium–calcium exchanger (NCX) was not expected to contaminate our signals. The obtained Ca$^{2+}$ decay was fitted by a mono-exponential function from 50% of the peak maximum to baseline resulting in the time constant $\tau$ as an indicator of decay velocity with r$^2$ > 0.93.

### Statistical analysis

Statistical analysis was performed using Prism software (GraphPad Software Inc). Data are presented as means ± SEM or means ±

SD. Numbers indicate the number of analyzed experiments and the number of cells isolated from at least three different animals. Statistically significant differences were determined either by paired or unpaired *t* test (two groups), by Mann–Whitney's test (if the datasets did not pass normality test), or by one-way ANOVA followed by a Dunnet's post-hoc test (multiple groups). $P < 0.05$ was considered as statistically significant. No hierarchical clustering of the datasets was observed after analysis according to Sikkel et al (2017).

## Supplementary Information

## Acknowledgements

We appreciate the excellent technical assistance of Birgit Schumann. The authors are grateful to Dr. T Murayama (Department of Pharmacology, Juntendo University School of Medicine, Tokyo, Japan) for the gift of HEK-293 RyR2 cells and HEK-293 RyR2 R-CEPIA*er* cells and to J Bienert (Facility for Synthetic Chemistry, MPI NAT), H Frauendorf (Georg-August-Universität Göttingen), and their co-workers. We highly appreciate the help of J Seikowski and J Schimpfhauser (Facility for Synthetic Chemistry, MPI NAT). We are grateful to L Cyganek (Stem Cell Unit, Department of Cardiology and Pulmonology, UMG Göttingen) for providing us with the RyR2-R2474S$^{+/-}$ iPSC cell line. This work was funded by the DZHK (Deutsches Zentrum für Herz-Kreislauf-Forschung, German Centre for Cardiovascular Research, 81Z0300117) to SE Lehnart, JW Wegener, GY Mitronova, T Pochechueva, L Ackermann, and G Hasenfuss. Additional funding was received by the Deutsche Forschungsgemeinschaft (DFG, German Research Foundation) under collaborative research center SFB1002 to GH (project D01), to SEL (projects A09 and S02); and under Germany's Excellence Strategy—EXC 2067/1- 390729940 to SE Lehnart.

### Author Contributions

JW Wegener: conceptualization, formal analysis, validation, investigation, visualization, and writing—original draft.
GY Mitronova: formal analysis, investigation, and writing—review and editing.
L ElShareif: formal analysis and investigation.
C Quentin: formal analysis and investigation.
V Belov: formal analysis and supervision.
T Pochechueva: investigation and cell culture.
G Hasenfuss: resources and project administration.
L Ackermann: resources.
SE Lehnart: conceptualization, data curation, funding acquisition, and writing—review and editing.

### Conflict of Interest Statement

SE Lehnart is an inventor on patent (US 20070089572A1) submitted by Columbia University "Novel agents for preventing and treating disorders involving modulation of RYR receptors."

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
