## [Reviewer comments · Life Science Alliance]

Life Science Alliance

A dual-targeted drug inhibits cardiac Ryanodine Receptor Ca²⁺-leak but activates SERCA2a Ca²⁺-uptake

Joerg Wegener, Gyuzel Mitronova, Lina ElShareif, Christine Quentin, Vladimir Belov, Tatiana Pochechueva, Gerd Hasenfuß, Lutz Ackermann, and Stephan Lehnart

DOI: <https://doi.org/10.26508/lsa.202302278>

Corresponding author(s): Joerg Wegener, Universitätsmedizin Göttingen; Stephan Lehnart, University of Göttingen Medical Center; and Joerg Wegener, Universitätsmedizin Göttingen

Review Timeline:

Submission Date:	2023-07-17
Editorial Decision:	2023-09-11
Revision Received:	2023-10-31
Editorial Decision:	2023-11-03
Revision Received:	2023-11-16
Accepted:	2023-11-17

Transaction Report:

September 11, 2023

Re: Life Science Alliance manuscript #LSA-2023-02278-T

Prof. Stephan E Lehnart
University of Göttingen Medical Center
Dept. of Cardiology
Heart Research Center Goettingen Dept. of Cardiology & Pulmonology
Göttingen, Niedersachsen D-37075
Germany

Dear Dr. Lehnart,

Thank you for submitting your manuscript entitled "A dual-targeted drug inhibits cardiac Ryanodine Receptor Ca²⁺-leak but activates SERCA2a Ca²⁺-uptake" to Life Science Alliance. The manuscript was assessed by expert reviewers, whose comments are appended to this letter. We invite you to submit a revised manuscript addressing the Reviewer comments.

Thank you for this interesting contribution to Life Science Alliance. We are looking forward to receiving your revised manuscript.

Sincerely,

B. MANUSCRIPT ORGANIZATION AND FORMATTING:

Reviewer #1 (Comments to the Authors (Required)):

The present manuscript of Wegener et al utilises a number of techniques to identify and test a number of compounds against Ryanodine Receptor leak and SR calcium uptake via SERCA. As a number of arrhythmogenic phenotypes are due to disturbances in calcium handling by these mechanisms, it is therefore of particular interest as compounds affecting these pathways could ultimately be novel antiarrhythmic agents.

In general the work has been carried out to a good standard, and the experiments have been well described. I do however, have a number of minor comments

- 1) In the introduction when CaMKii is being described, there should potentially be some mention of the EPAC pathways that also may contribute to SR leak.
- 2) At the end of page 4, the BSol domain is mentioned. I believe there needs to be a bit more information about this domain, as this is mentioned without any discussion.
- 3) When Istaroxime is mentioned, it should also be mentioned that this compound also has additionally mechanisms of action (eg inhibition of Na/K pump)
- 4) In the first paragraph of the results the term 'least variably' should be replaced by 'most consistently'.
- 5) Spelling error of 'Catecholamine' on page 11.
- 6) When i look at Figure 4, it appears the CaT decline for the novel compound could be biphasic. Also this representative example shows a slowing of the decline , just like is seen for dantrolene, whereas the majority of the data shows a faster decline. Is this just down to the choice of the 'representative' example, or is it to do with where you took the beginning of the decline from. Would it be worth trying to fit this to a biphasic curve?

Reviewer #2 (Comments to the Authors (Required)):

Summary

Heart failure and subsequent ca-mishandling is a major source of morbidity and mortality globally. The authors explored 1,4-benzothiazepine derivatives purported to combine activation of SR Ca²⁺ uptake with reduction of Ca²⁺ spark activity in cardiomyocytes. GM1869 was identified in creased ER Ca signal in HEK293 cells expressing RyR2 and HL-1 cells. GM1869 decreased Ca²⁺ spark frequency and slightly decreased FDHM in permeabilized cardiomyocytes. GM1869 reduced spark frequency and accelerated Ca re-uptake in RyR2R2474S cells and human hiPSC-CMs. Overall, the authors provide compelling evidence from multiple systems that GM1869 could improve Ca²⁺ handling in vivo for the treatment of heart failure. These data could be further supported by direct evidence of drug-target interactions.

Major Comments

The affinity of GM1869 is fairly low for clinical application, no?

What is with the images in Fig. 4C? Sample images of waves are not particularly clean. Also, the sample traces do not particularly reflect the data reported.

Evidence of direct drug binding at it's proposed targets, perhaps from a competition assay, would increase the impact of the manuscript.

Minor comments

The introduction is overly detailed and difficult to follow. It reads as more of a review of Ca handling rather than framing the specific question being addressed in the manuscript. A similar comment could be made regarding the discussion. Manuscript readability would improve from more concise writing.

Supplemental Figures would benefit from sample traces

Life Science Alliance manuscript #LSA-2023-02278-T**Response to reviewers****Response to reviewer 1**

Reviewer #1 (Comments to the Authors (Required)):

The present manuscript of Wegener et al utilises a number of techniques to identify and test a number of compounds against Ryanodine Receptor leak and SR calcium uptake via SERCA. As a number of arrhythmogenic phenotypes are due to disturbances in calcium handling by these mechanisms, it is therefore of particular interest as compounds affecting these pathways could ultimately be novel antiarrhythmic agents.

In general the work has been carried out to a good standard, and the experiments have been well described.

I do however, have a number of **minor comments**

1) In the introduction when CaMKii is being described, there should potentially be some mention of the EPAC pathways that also may contribute to SR leak.

Response to reviewer

The authors appreciate the reviewer's comment. The EPAC/NOS1 pathway described by (Pereira *et al*, 2017) has been included in the Introduction.

(p4: "RyR2 hyper-phosphorylation by Ca²⁺-calmodulin-dependent kinase (CaMKII) **involving exchange protein directly activated by cAMP (Epac2) and nitric oxide synthase 1 (NOS1)** (Pereira et al, 2017; Sag et al, 2009; Wehrens et al, 2004),")

2) At the end of page 4, the BSol domain is mentioned. I believe there needs to be a bit more information about this domain, as this is mentioned without any discussion.

Response to reviewer

The authors appreciate the reviewer's comment. The 4th paragraph at the end of page 4 mentioning the BSol domain has been moved to the Discussion according to the request of Reviewer 2 to shorten the Introduction. The BSol domain is explained there.

(p13: "In detail, ARM210 reversed the destabilization of the bridging solenoid domain (BSol, (Hadiatullah et al, 2022)) of RyR2 induced by the CPVT-associated RyR2-R2474S mutation thereby stabilizing the closed channel conformation (Miotto et al., 2022).")

3) When Istaroxime is mentioned, it should also be mentioned that this compound also has additionally mechanisms of action (eg inhibition of Na/K pump)

Response to reviewer

The authors feel sorry not to mention the dual-action of Istaroxime. We have included this in the manuscript.

(p4: "In contrast to the latter compound, that acts on SERCA2a and Na⁺/K⁺ ATPase (Micheletti *et al*, 2007), here").

4) In the first paragraph of the results the term 'least variably' should be replaced by 'most consistently'.

Response to reviewer

The authors appreciate the reviewer's comment. We have changed the term according to the reviewer.

5) Spelling error of 'Catecholamine' on page 11.

Response to reviewer

We have corrected the term according to the reviewer.

6) When i look at Figure 4, it appears the CaT decline for the novel compound could be biphasic. Also this representative example shows a slowing of the decline , just like is seen for dantrolene, whereas the majority of the data shows a faster decline. Is this just down to the choice of the 'representative' example, or is it to do with where you took the beginning of the decline from. Would it be worth trying to fit this to a biphasic curve?

Response to reviewer

The authors appreciate the reviewer's comments.

6.1. We checked whether the decay in Figure 4 C is biphasic. A fit of the data by a mono-exponential decay revealed tau values of 226 and 194 ms for control and GM1869, respectively (with $R^2 = 0.948$ and 0.943). The fit of the data by a bi-exponential decay revealed a fast tau of 102 and 81 ms for control and GM1869, respectively (with $R^2 = 0.953$ and 0.947). Thus, we confirmed the acceleration of the decay in both settings. However, in line with Occram's razor, we would like to stick to the smallest possible set of elements for analysis, i.e. a mono-exponential approach since it results already in reliable R^2 values > 0.9 .

6.2. We checked the representative example shown. A one-exponential fit of the data from 70% of the maximum to baseline resulted in tau values of 153 and 141 ms for control and GM1869, respectively, confirming an acceleration of the decay.

6.3. For further clarification, we have added the original data sets (means \pm SEM) into the inserts that were used for the respective mono-exponential fits.

Response to reviewer 2

Reviewer #2 (Comments to the Authors (Required)):

Summary

Heart failure and subsequent Ca²⁺-mishandling is a major source of morbidity and mortality globally. The authors explored 1,4-benzothiazepine derivatives purported to combine activation of SR Ca²⁺ uptake with reduction of Ca²⁺ spark activity in cardiomyocytes. GM1869 was identified in creased ER Ca signal in HEK293 cells expressing RyR2 and HL-1 cells. GM1869 decreased Ca²⁺ spark frequency and slightly decreased FDHM in permeabilized cardiomyocytes. GM1869 reduced spark frequency and accelerated Ca re-uptake in RyR2R2474S cells and human hiPSC-CMs. Overall, the authors provide compelling evidence from multiple systems that GM1869 could improve Ca²⁺ handling in vivo for the treatment of heart failure. These data could be further supported by direct evidence of drug-target interactions.

Major Comments

The affinity of GM1869 is fairly low for clinical application, no?

Response to reviewer

The authors appreciate the reviewer's concern. Indeed, we used maximal concentrations of the drugs (~0.1-10 μ M) to ascertain a reliable and comparable read-out of the drug effects. To get more insights towards the affinity of our compound we performed concentration-response curves for S36 and GM1869 in RyR2-WT and RyR2-R2474S^{+/−} iPSC-derived cardiomyocytes according to the comment of the reviewer. We measured spontaneous Calcium transients in control conditions and after application of the drug in a cumulative approach. We analyzed the drug effects on the rise time and the decay times of the transients. We found IC₅₀ values for S36 and GM1869 in the picomolar range. We have added these curves in the revised Figure 7. Thus, these data indicate that the affinity of GM1869 may be "fairly low" and sufficient for clinical application as an orally active drug.

What is with the images in Fig. 4C? Sample images of waves are not particularly clean. Also, the sample traces do not particularly reflect the data reported.

Response to reviewer

The authors appreciate the reviewer's comment.

(a) We acknowledge that the sample image is not particularly clean showing inhomogeneity in the shape of the wave that probably reflects inhomogeneity in some cellular structures of this particular cell after permeabilization. However, we took advantage of these inhomogeneity's as a tool to confirm that the region of interest has not been changed during the time course of the experiment, i.e. after drug application.

(b) The sample trace correspond to the upper left part of the wave where no inhomogeneity was observed and where the Calcium transient was analyzed in a 2 μm segment (as indicated in the Method section).

(c) We carefully checked the kinetics of decay in the sample traces in which the trace in the presence of the compound was slightly prolonged. However, a one-exponential fit of the data from 70% of the maximum to baseline resulted in tau values of 153 and 141 ms for control and GM1869, respectively, confirming an acceleration of the decay (see comment to point 6 of reviewer 1).

(d) We appreciate that the images displaying Calcium waves correspond to the Calcium transients shown and may not totally reflect the data shown in Suppl. Figure 4. However, we have included corresponding images into Suppl. Figure 4.

Evidence of direct drug binding at it's proposed targets, perhaps from a competition assay, would increases the impact of the manuscript.

Response to reviewer

The authors appreciate the reviewer's comment. We would be happy to perform competition assays with our new compound, especially since a competitor could serve as an antidote in an over-dosage setting. However, although the putative binding site for our new compound at the RyR2 may be similar to that described for ARM210, there is, to our knowledge, no drug described that can be used for a competition assay. We may have identified putative competitive compounds during our drug screening efforts but we focused on compounds that exhibited a modulatory effect on RyR2 activity. A re-screen of the combinations of the new compound with all of the identified inactive compounds may lead to the identification of a competitor but will be beyond the scope of this study and our sponsor.

Minor comments

The introduction is overly detailed and difficult to follow. It reads as more of a review of Ca handling rather than framing the specific question being addressed in the manuscript. A similar comment could be made regarding the discussion. Manuscript readability would improve from more concise writing.

Response to reviewer

The authors appreciate the reviewer's comment. We have shortened the introduction and applied a more concise writing.

Supplemental Figures would benefit from sample traces

Response to reviewer

The authors appreciate the reviewer's comment. We have added sample traces to Supplemental Figure 4.

References

Micheletti R, Palazzo F, Barassi P, Giacalone G, Ferrandi M, Schiavone A, Moro B, Parodi O, Ferrari P, Bianchi G (2007) Istaroxime, a stimulator of sarcoplasmic reticulum calcium adenosine triphosphatase isoform 2a activity, as a novel therapeutic approach to heart failure. *Am J Cardiol* 99: 24A-32A

Pereira L, Bare DJ, Galice S, Shannon TR, Bers DM (2017) beta-Adrenergic induced SR Ca(2+) leak is mediated by an Epac-NOS pathway. *J Mol Cell Cardiol* 108: 8-16

November 3, 2023

RE: Life Science Alliance Manuscript #LSA-2023-02278-TR

Prof. Stephan E Lehnart
University of Göttingen Medical Center
Dept. of Cardiology
Heart Research Center Goettingen
Göttingen, Niedersachsen D-37075
Germany

Dear Dr. Lehnart,

Thank you for submitting your revised manuscript entitled "A dual-targeted drug inhibits cardiac Ryanodine Receptor Ca²⁺-leak but activates SERCA2a Ca²⁺-uptake". We would be happy to publish your paper in Life Science Alliance pending final revisions necessary to meet our formatting guidelines.

- please add ORCID ID for the corresponding author--you should have received instructions on how to do so
- please add the Twitter handle of your host institute/organization as well as your own or/and one of the authors in our system
- please note that the titles in the system and on the manuscript file must match
- the full name (first name, middle name as initials, last name) of each author should be given on the title page
- please mark the corresponding Authors on the manuscript title page
- please consult our manuscript preparation guidelines <https://www.life-science-alliance.org/manuscript-prep> and make sure your manuscript sections are in the correct order
- we encourage you to revise the figure legend for figure S2 such that the figure panels are introduced in an alphabetical order
- please add callouts for Figures S2A-I and S4E-H to your main manuscript text
- the References in the Supplemental Material file should instead be incorporated into the main Reference list

A. FINAL FILES:

B. MANUSCRIPT ORGANIZATION AND FORMATTING:

Sincerely,

Point to point response to editorial comments

2023-11-03 15:58:15 2023-11-03 15:58:15 Redacted Redacted Life Science Alliance Manuscript - Editorial Decision LSA-2023-02278-TR November 3, 2023 RE: Life Science Alliance Manuscript #LSA-2023-02278-TR Prof. Stephan E Lehnart University of Göttingen Medical Center Dept. of Cardiology Heart Research Center Goettingen Göttingen, Niedersachsen D-37075 Germany Dear Dr. Lehnart, Thank you for submitting your revised manuscript entitled "A dual-targeted drug inhibits cardiac Ryanodine Receptor Ca²⁺-leak but activates SERCA2a Ca²⁺-uptake". We would be happy to publish your paper in Life Science Alliance pending final revisions necessary to meet our formatting guidelines. Along with points mentioned below, please tend to the following:  -please add ORCID ID for the corresponding author--you should have received instructions on how to do so -please add the Twitter handle of your host institute/organization as well as your own or/and one of the authors in our system -please note that the titles in the system and on the manuscript file must match -the full name (first name, middle name as initials, last name) of each author should be given on the title page -please mark the corresponding Authors on the manuscript title page -please consult our manuscript preparation guidelines https://www.life-science-alliance.org/manuscript-prep and make sure your manuscript sections are in the correct order 	Corrected Not applicable corrected corrected corrected corrected corrected
---	---

-we encourage you to revise the figure legend for figure S2 such that the figure panels are introduced in an alphabetical order -please add callouts for Figures S2A-I and S4E-H to your main manuscript text -the References in the Supplemental Material file should instead be incorporated into the main Reference list If you are planning a press release on your work, please inform us immediately to allow informing our production team and scheduling a release date. LSA now encourages authors to provide a 30-60 second video where the study is briefly explained. We will use these videos on social media to promote the published paper and the presenting author (for examples, see https://twitter.com/LSAJournal/timelines/1437405065917124608). Corresponding or first-authors are welcome to submit the video. Please submit only one video per manuscript. The video can be emailed to contact@life-science-alliance.org To upload the final version of your manuscript, please log in to your account: https://lsa.msubmit.net/cgi-bin/main.plex You will be guided to complete the submission of your revised manuscript and to fill in all necessary information. Please get in touch in case you do not know or remember your login name. To avoid unnecessary delays in the acceptance and publication of your paper, please read the following information carefully. A. FINAL FILES: These items are required for acceptance. -- An editable version of the final text (.DOC or .DOCX) is needed for copyediting (no PDFs). -- High-resolution figure, supplementary figure and video files uploaded as individual files: See our detailed guidelines for preparing your production-ready images, https://www.life-science-alliance.org/authors -- Summary blurb (enter in submission system): A short text summarizing in a single sentence the study (max. 200 characters including spaces). This text is used in conjunction with the titles of papers, hence should be informative and complementary to the title. It should describe the context and significance of the findings for a general readership; it should be written in the present tense and refer to the work in the third person. Author names should not be mentioned.	corrected corrected corrected Not applicable Not applicable corrected yes yes, >300dpi yes
--	--

B. MANUSCRIPT ORGANIZATION AND FORMATTING: Full guidelines are available on our Instructions for Authors page, https://www.life-science-alliance.org/authors We encourage our authors to provide original source data, particularly uncropped/-processed electrophoretic blots and spreadsheets for the main figures of the manuscript. If you would like to add source data, we would welcome one PDF/Excel-file per figure for this information. These files will be linked online as supplementary "Source Data" files. **Submission of a paper that does not conform to Life Science Alliance guidelines will delay the acceptance of your manuscript.** **It is Life Science Alliance policy that if requested, original data images must be made available to the editors. Failure to provide original images upon request will result in unavoidable delays in publication. Please ensure that you have access to all original data images prior to final submission.** **The license to publish form must be signed before your manuscript can be sent to production. A link to the electronic license to publish form will be available to the corresponding author only. Please take a moment to check your funder requirements.** **Reviews, decision letters, and point-by-point responses associated with peer-review at Life Science Alliance will be published online, alongside the manuscript. If you do want to opt out of having the reviewer reports and your point-by-point responses displayed, please let us know immediately.** Thank you for your attention to these final processing requirements. Please revise and format the manuscript and upload materials within 7 days. Thank you for this interesting contribution, we look forward to publishing your paper in Life Science Alliance. Sincerely, Eric Sawey, PhD Executive Editor Life Science Alliance http://www.lsjournal.org	yes Not applicable Not applicable yes yes yes yes yes
--	---

November 17, 2023

RE: Life Science Alliance Manuscript #LSA-2023-02278-TRR

Dr. Joerg W Wegener
Universitätsmedizin Göttingen
Cardiology & Pneumology
Robert Koch 42
Goettingen, Germany 37075
Germany

Dear Dr. Wegener,

Thank you for submitting your Research Article entitled "A dual-targeted drug inhibits cardiac Ryanodine Receptor Ca²⁺-leak but activates SERCA2a Ca²⁺-uptake". It is a pleasure to let you know that your manuscript is now accepted for publication in Life Science Alliance. Congratulations on this interesting work.

DISTRIBUTION OF MATERIALS:

Again, congratulations on a very nice paper. I hope you found the review process to be constructive and are pleased with how the manuscript was handled editorially. We look forward to future exciting submissions from your lab.

Sincerely,
